# High sensitivity cameras can lower spatial resolution in high-resolution optical microscopy

Henning Ortkrass [1] ✉, Marcel Müller[1], Anders Kokkvoll Engdahl[1], Gerhard Holst[2] & Thomas Huser [1] ✉

High-resolution optical fluorescence microscopies and, in particular, super-resolution fluorescence microscopy, are rapidly adopting highly sensitive cameras as their preferred photodetectors. Camera-based parallel detection facilitates high-speed live cell imaging with the highest spatial resolution. Here, we show that the drive to use ever more sensitive, photon-counting image sensors in cameras can, however, have detrimental effects on the spatial resolution of the resulting images. This is particularly noticeable in applications that demand a high space-bandwidth product, where the image magnification is close to the Nyquist sampling limit of the sensor. Most scientists will often select image sensors based on parameters such as pixel size, quantum efficiency, signal-to-noise performance, dynamic range, and frame rate of the sensor. A parameter that is, however, typically overlooked is the sensor's modulation transfer function (MTF). We have determined the wavelength-specific MTF of front- and back-illuminated image sensors and evaluated how it affects the spatial resolution that can be achieved in high-resolution fluorescence microscopy modalities. We find significant differences in image sensor performance that cause the resulting spatial resolution to vary by up to 28%. This result shows that the choice of image sensor has a significant impact on the imaging performance of all camera-based optical microscopy modalities.

The availability and application of optical microscopy based on image sensors is rapidly increasing because of the high read-out rates combined with the high quantum efficiency of modern scientific complementary metal-oxide semiconductor (sCMOS) based image sensors. The introduction of sCMOS sensors has also lowered the cost of ownership of single photon counting camera systems. High-speed readout of sCMOS cameras is paramount for advanced high-resolution live cell imaging, such as lattice light-sheet microscopy[1], single-objective light sheet microscopy[2,3], and oblique plane structured illumination microscopy[4]. sCMOS sensors employing rolling-shutter readout are now frequently being exploited in order to enable line-confocal out-of-focus signal rejection in novel light-sheet and super-resolution microscopy modalities, where the rolling shutter acts as a virtual pinhole slit[5,6]. These image sensors have further found rapid adoption in the majority of super-resolution imaging modalities, such as super-resolution structured illumination microscopy (SR-SIM)[7,8], single molecule localization microscopy (SMLM)[9,10], and signal fluctuation-based super-resolution microscopies (e.g. SOFI)[11], to name a few. Arguably, their most widespread use is, however, in standard wide-field fluorescence microscopy, where their performance is essential, especially in high-resolution imaging applications[12–14].

[1]Biomolecular Photonics, Faculty of Physics, Bielefeld University, Bielefeld, Germany. [2]Excelitas PCO GmbH, Donaupark 11, Kelheim, Germany. ✉e-mail: hortkrass@physik.uni-bielefeld.de; thomas.huser@physik.uni-bielefeld.de

sCMOS sensors can be realized in both front- and back-illuminated variants, and with different semi-conductor parameters, e.g. resistance load. In front illumination, the electronic circuitry (transistors and conduits) are structured onto the light-sensitive front side of the sensor (the negative influence of the poor fill factor that results in many cases from this architecture is often partially compensated by the addition of micro lenses in front of each sensor pixel). In back-illuminated sensors, these structures are on the opposite side of the light-sensitive surface. In general, this leads to a lower quantum efficiency (QE) of front-illuminated sensors, as by design some surface area is covered by the electronics and is not light sensitive. QE is often and rightfully seen as an important quality factor describing the sensitivity of an image sensor when choosing a camera for fluorescence microscopy applications, thus back-illuminated sensors typically appear to be the superior choice for these systems.

Back-illumination of images sensors does, however, come with a drawback: once photons are converted to photo-electrons in the doped silicon, they have to traverse a much longer path through the thinned silicon to reach the potential well of the pixel where they are collected. This increases the possibility of electrons being scattered into neighboring pixels, an effect called pixel crosstalk[15]. While in principle known, this effect is often not accounted for, and most datasheets quote QE as a key sensor parameter, but they do not provide the actual sensor's modulation transfer function (MTF) that can be used to quantify this effect.

The incoherent wide-field detection scheme of the optical system of a fluorescence microscope itself, which is common to general wide-field imaging, such as wide-field fluorescence, SR-SIM, light-sheet microscopy, and spinning disk confocal microscopy, to name a few, has a MTF that falls off steeply towards higher spatial frequencies[16]. In a high-resolution optical system, where the magnification is chosen to achieve a projected pixel size that fulfills the Nyquist sampling criterion, the question arises whether or not a change in the camera sensor's MTF actually contributes significantly to the overall resolution. And, if it does, will the higher QE provided by the back-illuminated sensors compensate for their lack in high-frequency response?

## Results and Discussion
### Modulation-transfer function (MTF) of different image sensors with the same pixel size

We investigated the effect of, in total, 4 different image sensors, two front- (C1 and C2) and two back-illuminated sCMOS sensors (C3 and C4), with the same pixel size of 6.5 μm x 6.5 μm and the same sensor size of 2048 × 2048 pixel (each in different camera implementations, see Table 1) by determining their performance with wide field fluorescence and two popular super-resolution imaging methods, SR-SIM and direct stochastic optical reconstruction microscopy (dSTORM)[17]. In addition, two more front-illuminated sCMOS cameras from different manufacturers were also evaluated (C5 and C6, see Table 1). A custom-constructed microscope, capable of high-resolution wide-field as well as SR-SIM imaging, was used for this purpose. Wide field fluorescence images were acquired with a f = 180 mm tube lens (overall magnification 60x) and a f = 250 mm tube lens (overall magnification 83.3x) to measure the effect of the sensor MTF at two projected pixel sizes of 108 nm and 78 nm[18]. The Nyquist criterion for a nominally 60x (f = 3 mm) 1.5NA objective lens at green emission wavelengths is only fulfilled with the f = 250 mm tube lens and an overall magnification of 83.3x. The super-resolution imaging was performed with a f = 250 mm tube lens, thus 78 nm projected pixel size and the fluorescence signal was split by a 50/50 beam splitter and focused on the two image sensors by identical tube lenses. Electronic synchronization allowed the two cameras to receive frame-by-frame identical data.

To quantify the effect of the MTF on super-resolution fluorescence microscopy images we first measured the overall MTF of the microscope for all image sensors. Single 100 nm TetraSpeck (TS)

beads (Thermo Fisher Scientific) were imaged in focus in the center of the field-of-view (FOV) ten times each, utilizing almost the maximum dynamic range of each camera. This was repeated with several different beads. The contrast of the images was maximized by subtracting the mean of the background signal and the image stacks were Fourier-transformed, deconvolved with the lateral projected spherical bead shape, and averaged[7]. The two-dimensional MTF was azimuthally averaged and set to zero outside its (theoretically calculated) known support. The MTF was measured at 555 nm and 665 nm emission wavelengths for front-illuminated (FSI) and back-illuminated (BSI) sensors. We confirmed that both, the transmitted and reflected image path after the beam splitter cube exhibited the same photon count rate and MTF (see Supplementary Information). In a well-aligned microscope setup, it is typically assumed that the MTF mainly depends on the objective lens and the camera sensor - depending on the projected pixel size. We find, however, that the MTF is different for all sensor types, especially for spatial frequencies higher than 1/μm at a projected pixel size of 78 nm. The MTF of the BSI sensor C4 is, e.g. 24% lower than the MTF of the BSI sensor C3 at 665 nm and at a spatial frequency of 2/μm. This is explained by pixel crosstalk, which becomes more and more significant at higher spatial frequencies. The cutoff frequencies remain the same for all sensor types as these are only dependent on the numerical aperture of the objective lens. The actual image resolution, however, also depends on the magnitude of the MTF because Poisson noise limits the spectrum of the actually detectable spatial frequencies and, in general, contrast. MTF data acquired by imaging fluorescent beads are considered as data acquired in a low-photon count regime. To further explore the influence of measurements taken in a high-photon count regime, noise, and alternative incoherent light sources on the overall MTF we also acquired MTF data by illuminating a 200 nm diameter hole in an otherwise opaque aluminum film. Here, a 200 nm diameter hole was milled by focused ion-beam milling into a 100 nm thick aluminum layer on a 170 μm thick cover glass. The hole was positioned in the center of the field of view and then illuminated with a white light LED from behind. It was imaged onto a single pixel with a 1.4 NA, 40x objective lens, and 11.8x overall magnification.

To examine the effect of the different MTFs on wide field and super-resolved fluorescence images, we tested sensor C1 compared to C4 for wide field and SR-SIM, since they displayed the lowest and highest MTFs at 555 nm, and sensors C2 and C4 for dSTORM.

### Wide field imaging with 60x and 83.3x magnification

The camera sensor performance was first evaluated by imaging the membrane of fluorescently stained liver sinusoidal endothelial cells (LSEC) with different magnifications and with the same optical setup (see Fig. 1). The magnification was changed by using different tube lenses with the same 60×1.5NA Olympus objective lens. We used a standard 180 mm Olympus tube lens and a Ploessel type 250 mm tube lens to achieve fluorescence detection, the Abbe resolution limit is 171 nm, corresponding to a sampling of 1.58x at 60x magnification and a sampling of 2.19x at 83.3x magnification (calculated for green/eGPF emission). Note that the 60x/180 mm objective lens/tube lens combination does not achieve enough magnification for Nyquist sampling with high-NA objective lenses and current sCMOS pixel sizes (6.5 μm). Still, as this combination is readily available, it is used in a very large number of optical microscopes around the world. We measured the actual resolution achieved with the two sensors for these magnifications and at green emission wavelengths by calculating Fourier ring correlation (FRC)[19] data on the wide field images. FRC is a method originally introduced in electron microscopy in order to measure the correlation for 2 images at different spatial frequencies. For this purpose, two images of the same sample have to be acquired and are then correlated against each other. If the correlation coefficient drops below 1/7 (0.14), the signal is dominated by noise and the resolution cutoff is reached. We use this value throughout the paper to compare

the spatial resolution that can be reached with different image sensors with otherwise identical optical systems. For the 60x magnification, five different FOVs were imaged ten times each, and the individual FRCs were averaged (Fig. 1c). With this modality, the images for the sensors were acquired sequentially at identical conditions. The front-illuminated sensor C1 gives a resolution limit of 242 nm and the back-illuminated sensor C4 provides 310 nm resolution. With the 83.3x magnification, two different FOVs were imaged 30 times on both sensors, simultaneously, and the averaged FRC (Fig. 1d) shows a resolution limit of 220 nm for C1 and 260 nm for C4. It should also be noted that the samples imaged in wide-field fluorescence and SR-SIM modalities all exhibited high fluorescence signal levels. This result indicates that the image resolution is severely limited by the sensor

MTF with a standard tube lens, and the resolution limit heavily depends on the sensor type. The effect is strong enough to not only be picked up by quantitative measurements such as FRC, but is also clearly visible by eye when comparing fine structural details (see Fig. 1a, b magnified insets II vs IV).

With a magnification of 83.3x, the effect becomes less significant. This is to be expected, as the finer sampling (2.19x instead of 1.58x) spreads the PSF across more pixels, so the effect of the sensor MTF becomes less pronounced in comparison. For super-resolution imaging, we thus chose the 83.3x magnification with the 250 mm tube lens, corresponding to a pixel size similar to commercial SR-SIM setups, where typically some oversampling is employed.

### Super-resolution structured illumination microscopy (SR-SIM)

The comparison of the camera sensors with respect to their performance in SR-SIM was tested on LSECs at 488 nm (stained against the actin cytoskeleton) and 640 nm (plasma membrane stain) excitation wavelengths in 2D- and total internal reflection fluorescence (TIRF) SIM mode, which results in different spatial resolutions[18]. Raw images were acquired simultaneously on both cameras and reconstructed with the same parameter set and the sensor-dependent MTF. The image reconstruction was performed using the open access fairSIM plugin in ImageJ (v1.54j)[20,21].

With a resolution improvement of ~2x, which is typical for TIRF-SIM, we found a significant difference in the resolution limit for actin structures excited with 488 nm and imaged at 505 nm wavelength. The spatial resolution that we achieved with the C1 sensor (as measured by the FRC) is 85 nm, while the resolution limit of the images acquired with the C4 sensor is 93 nm which corresponds to a difference in

**Table 1 | The camera abbreviations correspond to the noted camera and sensor models**

|  | Camera model | Sensor type |
|---|---|---|
| C1 | pco.edge 4.2 | FSI image sensor: CIS2020AF |
| C2 | pco.panda 4.2 | FSI image sensor: GSENSE2020 |
| C3 | pco.edge 4.2 bi | BSI image sensor: GSENSENE2020BSI-H |
| C4 | pco.panda 4.2 bi | BSI image sensor: GSENSE2020BSI-M |
| C5 | Andor Neo 5.5 | FSI image sensor: CIS2051A / CIS2521 (the original naming convention by Fairchild Imaging later BAE Fairchild Imaging later BAE Systems) has changed in this time) |
| C6 | Hamamatsu Orca Flash 4.0 | FSI image sensor: CIS2020 or CIS2020A |

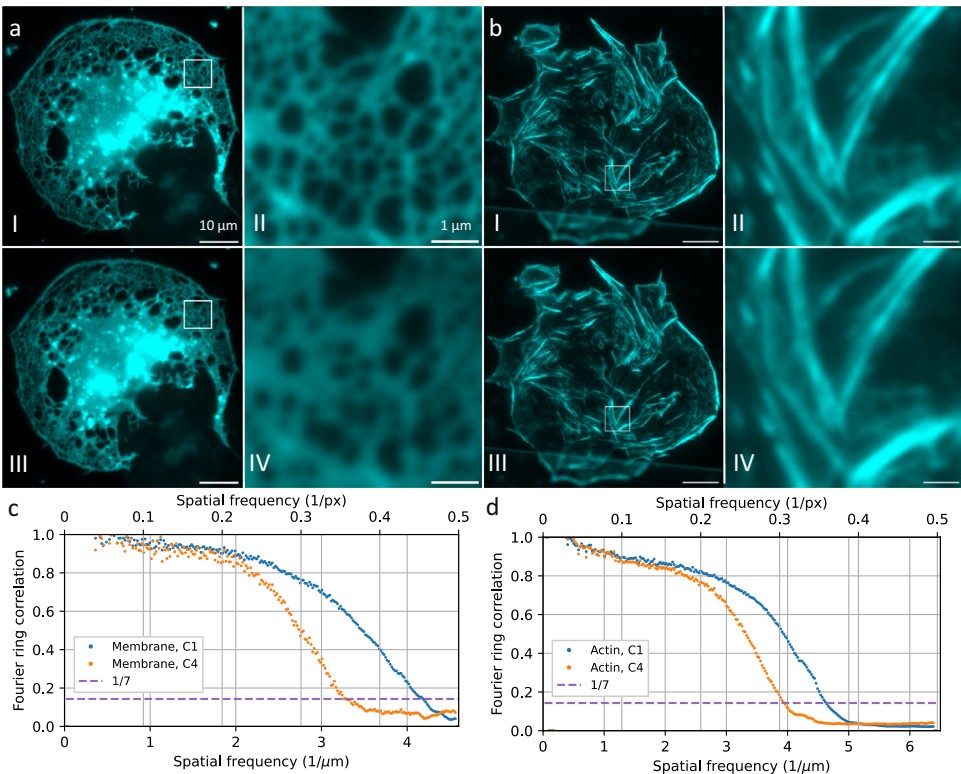

**Fig. 1 | Wide-field images of liver sinusoidal endothelial cells stained with Vybrant DiO (a) and Phalloidin AF488 (b), both with green emission.** The cells are imaged sequentially (**a**) or simultaneously (**b**) with the C1 sensor (aI, bI) and C4 sensor (aIII, bIII). To investigate the effect of the sensor MTF, different tube lenses with different magnifications were used. a was acquired with a total magnification of 60x, b with 83.3x. The images show a significant difference in resolution and contrast between the different sensors. The Fourier ring correlation (**c**) shows a sensor limited resolution limit of 242 nm for sensor C1 and 310 nm for C4 with 60x magnification. It is calculated from the FRC average of six different FOV with 10 frames each. The FRC for the images with a 83.3x magnification (**d**) shows a resolution limit of 220 nm for C1 and 260 nm for C4. It is the average of the FRC of two FOV with 30 frames each. Scale bar is 10 μm (aI, aIII, bI, bIII) and 1 μm (aII, aIV, bII, bIV). The images a and b show a single acquisition.

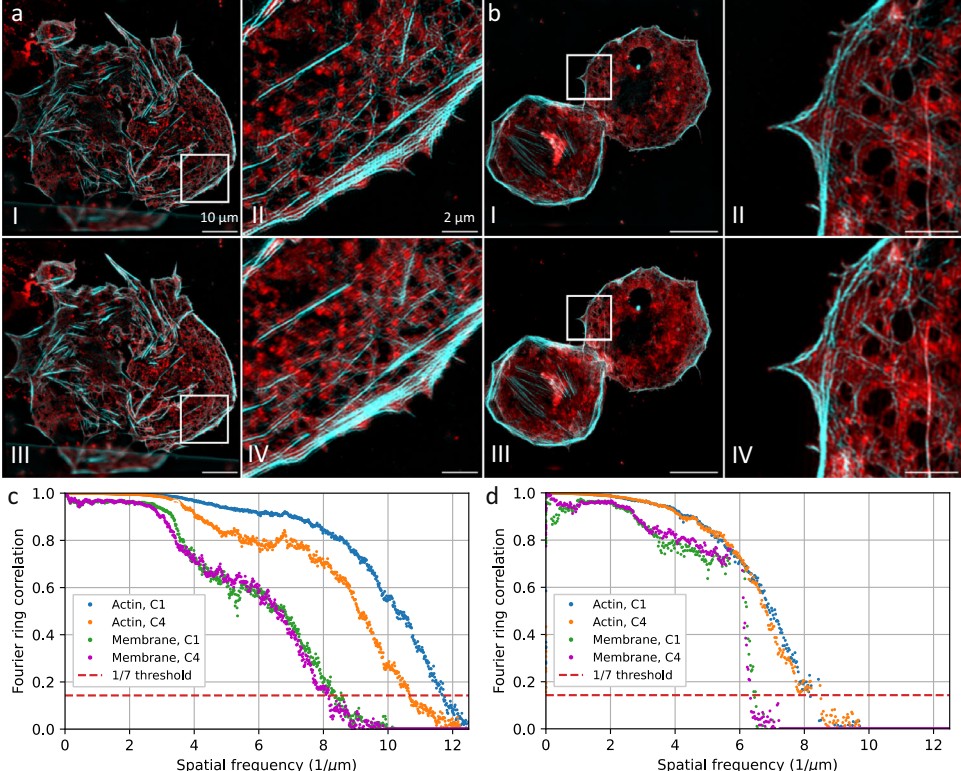

**Fig. 2 | Reconstructed SR-SIM images of liver sinusoidal endothelial cells, stained with Phalloidin AF488 and membrane dye Biotracker 647.** The cells are imaged simultaneously by the C1 sensor (aI, bI) and the C4 sensor (aIII, bIII) in TIRF (**a**) and 2D-SIM (**b**) mode. Fourier ring correlation (FRC) analysis reveals the different resolution improvements for TIRF-SIM (**c**) and 2D-SIM (**d**) for both sensor types and both wavelengths. The resolution limit corresponding to the frequency cutoff is significantly different for the actin cytoskeleton (shown in cyan) imaged by excitation at 488 nm with TIRF-SIM: 85 nm with the C1 sensor and 93 nm with the C4 sensor. The FRC curves vary also for the plasma membrane (excited at 647 nm) imaged in TIRF and 2D-SIM mode, but at red wavelengths, the resolution limit varies less than 2% for the sensor types. Scale bar is 10 µm (aI, aIII, bI, bIII) and 2 µm (aII, aIV, bII, bIV). The images a and b show a single acquisition.

resolution of 8% between the two different sensors. We explain this result based on the high-frequency cutoff in the raw images of 4.9/µm which corresponds to 0.38/px and is close to the Nyquist sampling limit of 0.5/px and therefore most sensitive to pixel crosstalk. At lower frequency cutoffs, as obtained for the red-emitting plasma membrane stain, the FRC varies by less than 2%. Images acquired in 2D-SIM mode (resulting in a resolution improvement of ~1.7x) show differences in the resolution limit between FSI and BSI sensors of less than 1% (Fig. 2). This is explained by the lower spatial frequency of the excitation pattern, that causes a modulation pattern in the image data that is nearly unaffected by pixel crosstalk.

**Direct stochastic optical reconstruction microscopy (*d*STORM)**
In order to investigate the influence of MTF's of the different sensor types on single molecule localization microscopy (SMLM), we performed *d*STORM on immunofluorescently labeled microtubules in U2OS cells. The same microscope setup, in particular the same detection scheme, was used as in the SR-SIM experiments. However, instead of SIM patterns, classic wide-field fluorescence excitation at 647 nm wavelength was used with the fluorescence signal again split 50/50 to the two sCMOS image sensors. As can be seen in Fig. 3, the *d*STORM reconstruction of 10,000 camera frames exhibits only a slight difference in spatial resolution between data acquired by the different image sensors. This is expected based on the wavelength-specific response of the sensor and further exacerbated by the *d*STORM image reconstruction process, where the image is composed of points representing the centroid of a 2D fit function for each molecule detected. The precision of localization scales with the PSF width, which is somewhat influenced by the sensor MTF, but

also with the square root of the number of detected photons, which of course benefits from an increase in sensor quantum efficiency. The effect of the MTF on *d*STORM localization precision therefore appears to be negligible for this type of super-resolution microscopy.

We have demonstrated that the image sensor type has a significant impact on a microscope's MTF and, therefore, significantly affects the spatial resolution in wide field fluorescence imaging close to the Nyquist limit.

At 60x magnification - a very popular choice for many commercially available implementations of high-NA, high-resolution wide-field, and super-resolution fluorescence imaging - the effect significantly reduces the image quality. A similar impact should be expected for any other high-resolution imaging task where pixel size is kept close to or even below the Nyquist criterion. This effect could potentially be compensated by a change in magnification based on the sensor type, but this is often difficult to realize in practice.

At 83.3x magnification, a choice providing some Nyquist over-sampling (typical for commercial high- and super-resolution systems), the effect is less pronounced but still clearly noticeable in both wide-field and SR-SIM imaging modalities. For *d*STORM, as an example of SMLM, however, it is not significant, as the reliance on molecule localization and their dependence on detected photons compensates for the reduced MTF.

The MTFs of the image sensors differ significantly for spatial frequencies in the image plane above 0.1/pixel. We explain this effect with the underlying sensor architecture, which affects the likelihood of pixel crosstalk. This affects both wide-field imaging as well as TIRF-SIM reconstructions at emission wavelengths in the green part of the

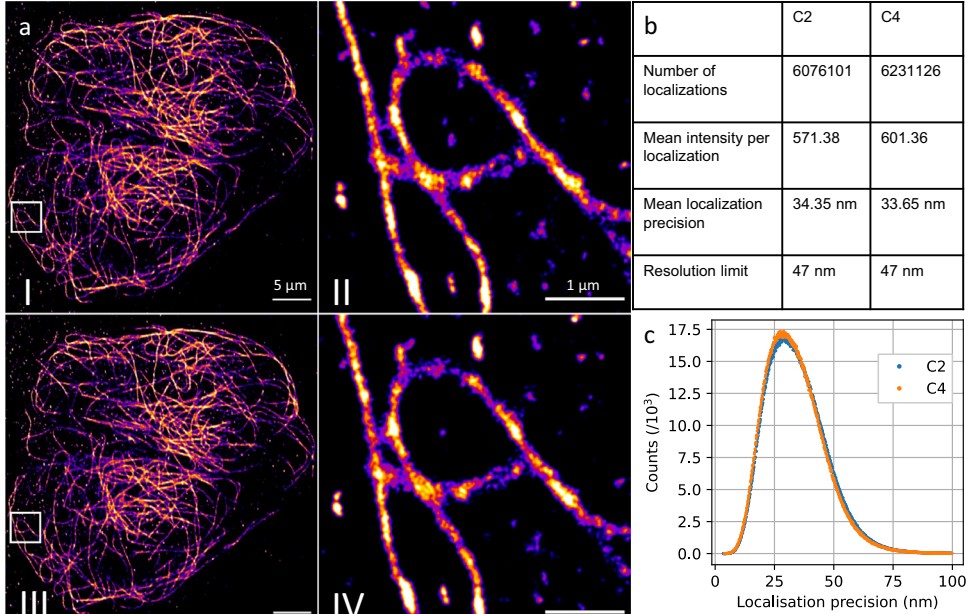

| b | C2 | C4 |
|---|---|---|
| Number of localizations | 6076101 | 6231126 |
| Mean intensity per localization | 571.38 | 601.36 |
| Mean localization precision | 34.35 nm | 33.65 nm |
| Resolution limit | 47 nm | 47 nm |

**Fig. 3 | Super-resolution direct stochastic optical reconstruction microscopy images of Alexa647-immunofluorescently stained microtubuli in an U2OS cell (a).** (aI, aII): reconstruction of the raw data acquired by the C2 sensor, (aIII, aIV): simultaneously acquired images using the C4 sensor, reconstructed with the same parameters. The mean localization precision (b, c) and resolution is similar for both cameras, despite the higher photon count per localization obtained with the back-illuminated sensor. Scale bars are 5 μm (aI), (aIII) and 1 μm (aII), (aIV). The experiment was conducted once.

visible spectrum with a frequency cutoff close to the Nyquist sampling rate. The MTF directly impacts the signal-to-noise ratio of high spatial frequencies of the reconstructed image and thus the resolution limit. Here, sample structures imaged at green wavelengths exhibit a difference in spatial resolution of up to 28%. In summary, the MTF of the image sensor plays a critical role in the ultimate spatial resolution that can be achieved in high resolution microscopy with modern sCMOS image sensors. In order to avoid this detrimental effect on a microscope's spatial resolution, image data should be acquired significantly above the Nyquist sampling limit (depending on the application by up to 30–50% above the Nyquist limit). This result also impacts other types of camera-based high resolution microscopy modalities, such as spinning-disk confocal, and light-sheet fluorescence. Depending on the application it should be carefully evaluated which sensor type is chosen, because the sensor type (FSI vs. BSI) can play a more critical role than the quantum efficiency of the sensor.

## Methods
### Microscope Setup
The MTF measurements and SIM-imaging experiments were performed with a wide-field microscope with a fiber-based 2D-SIM excitation path. We used a 491 nm laser for excitation (Cobolt Calypso 100), as well as a 532 nm laser (Coherent Compass 215M-50) and a 639 nm laser (Photontec MSL-FN-639-300). The excitation pattern was projected into the sample with an Olympus 60×1.5NA objective lens (UPLAPO60XOHR) and the fluorescence was epi-detected, split by a 50/50 non-polarizing beam splitter cube (Qioptiq G335525000) and imaged either by a Ploessel-type tube lens (constructed out of two Thorlabs achromatic lenses AC508-500-A) with f = 250 mm or a f = 180 mm tube lens (Olympus U-SWATLU) onto the image sensors. The difference in the MTF between both imaging paths was checked to be negligible. We also frequently swapped cameras and took measurements with cameras in each beam path to ensure that MTF values were not affected by a specific beam path. The projected pixel size was 78 nm with the Ploessel-type tube lens or 108 nm with the Olympus tube lens.

### Modulation Transfer Function (MTF) measurement
For the calculation of the modulation transfer function (MTF) for different sensors, we imaged a test sample in bright field mode with the different camera models. The test sample consisted of 200 nm diameter transparent holes that were milled into an opaque aluminum layer by focused ion beam milling. The aluminum layer was sputter-coated onto a cover glass with a thickness of 100 nm and a 5 nm thick chromium adhesion layer. The hole pattern was illuminated by a collimated beam of white LED light. The pattern was imaged by an Olympus 40×1.4 NA objective lens (UPLXAPO40XO) and a Ploessel-type f = 53 mm tube lens onto the camera sensor, resulting in a total magnification of 11.8x. With an optical resolution of approximately 200 nm, the size of the bright holes of the test sample on the sensor is well below the projected pixel size of 550 nm for all camera sensors. For measuring the sensor MTF, the image of one hole was precisely focused and centered onto one sensor pixel and imaged >10 times with an intensity of typically 0.7 of the dynamic bandwidth of the sensor. The images were Fourier transformed and averaged azimuthally and across the entire stack. The MTFs for the cameras C1, C4, C5, and C6 are shown in Fig. 4, as well as their corresponding PSFs. This corresponds to 4 different sCMOS cameras from 3 different manufacturers.

### Cell preparation
U2OS cells[22] were seeded on #1.5 glass coverslips. For indirect antibody-staining of tubulin filaments, the U2OS cells were fixed with 4% paraformaldehyde for 10 minutes at room temperature. An extraction step was followed by fixation with 0.5% glutaraldehyde in PEM buffer (PEM: 80 mM piperazine-N,N-bis(2-ethanesulfonic acid) (PIPES), 5 mM egtazic acid (EGTA), 2 mM $MgCl_2$ at pH 6.8). After washing with phophate buffered saline (PBS), glutaraldehyde-induced autofluorescence was quenched by the addition of 0.1% $NaBH_4$ in PBS for 7 minutes followed by washing with PBS three times. Cells were blocked and permeabilized using a blocking buffer containing 0.3% gelatin and 0.05% Triton X-100 in PBS for 1 hour. Fluorescence staining was performed overnight at 4 °C for $\alpha$- and $\beta$-tubulin using a mixture of three primary antibodies (T5168, T6199, T5923, Sigma) at a combined

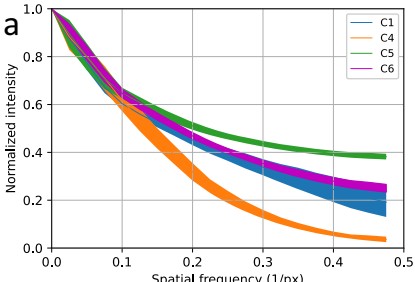
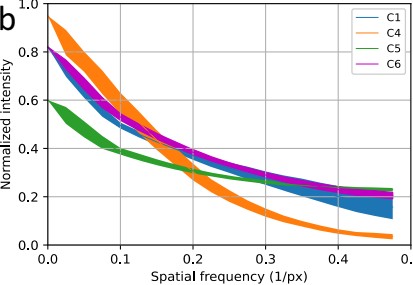
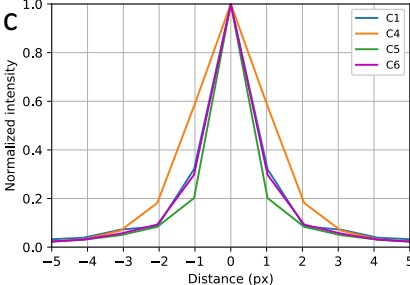

**Fig. 4 | Comparison of the modulation transfer function (MTF) of the front-illuminated (FSI) and back-illuminated (BSI) image sensors.** The normalized MTFs (**a**) differ significantly between the sensors in the high frequency regime. When scaled with the quantum efficiencies of the sensors (**b**), the differences are still significant. The line width corresponds to the standard deviation. The normalized PSFs corresponding to the MTFs are shown in **c**. The curves are derived from more than ten experiments.

dilution of 1:150 in blocking buffer. Cells were washed with PBS three times and the second staining solution, AF647-conjugated goat anti-mouse IgG secondary antibody (A-21237, ThermoFisher) diluted 1:200 in blocking buffer, was incubated at room temperature for 90 minutes. For dSTORM measurements the common GODCAT buffer containing the enzymatic oxygen scavengers glucose oxidase and catalase, with beta-mercaptoethanol (BME) as a switching agent was used to induce intermittent fluorescence of the AF647 fluorophores.

Cryo-preserved rat liver sinusoidal endothelial cells (LSECs) were a kind gift of Dr. Peter McCourt and Dr. Karolina Szafranska at UiT - the Arctic University of Norway. They were shipped on dry ice to Germany and stored at −80 °C. For thawing and seeding, a vial with LSECs was placed in an incubator at 37 °C until nearly all the ice had thawed. The cells were gently pipetted drop-wise to 25 ml of pre-warmed Dulbecco's Modified Eagle Medium (DMEM) and centrifuged at $50\,g$ for 3 minutes to remove any hepatocytes remaining from the cell isolation. The supernatant containing LSECs was used for a second centrifugation step at 300 g for 8 minutes. The cell pellet was resuspended in 4 ml–7 ml DMEM and 1.5 ml (~100,000 cells per cm²) of the cell solution was pipetted onto a fibronectin-coated #1.5 glass coverslip. The coverslip surface was coated with fibronectin (0.2 mg/ml) in phosphate buffered saline (PBS) containing 2 mM ethylenediaminetetraacetic acid (EDTA) for 1 hour at room temperature and washed with PBS afterward. After allowing the cell suspension to incubate on the glass coverslip for 1 hour at 37 °C and 5% CO₂, the coverslip was washed with pre-warmed DMEM and incubated for another 2 hours before fixation with 4% formaldehyde in PBS for 10 minutes at room temperature. First, the plasma cell membrane was stained with BioTracker 655 Red Cytoplasmic Membrane Dye (SCT108, Sigma) diluted 1:200 in PBS for 1 hour at room temperature. The cells were washed twice in PBS before staining the actin cytoskeleton. The LSECs were incubated in a 1:40 dilution of Phalloidin CF568 (00044-T, Biotium) in PBS for 2 hours at room temperature. After the staining process was completed, the cells were washed 3 times with PBS.

### Reporting summary

Further information on research design is available in the Nature Portfolio Reporting Summary linked to this article.

## Data availability

The MTF and image data generated in this study have been deposited in the figshare database under accession code https://doi.org/10.6084/m9.figshare.25213490.

## Code availability

The custom software code used to derive the MTF from the image data has been deposited in the figshare database under accession code https://doi.org/10.6084/m9.figshare.25213490.

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

## Acknowledgements

The authors thank Prof. Peter McCourt and Dr. Karolina Szafranska (UiT - the Arctic University of Norway) for the kind gift of rat liver sinusoidal endothelial cells. They also thank Jasmin Schürstedt-Seher and Dr. Wolfgang Hübner for the sample preparation. This work received funding from the European Union's European Innovation Council PATHFINDER Open Programme under grant agreement No 101046928. The project was also supported by the German Federal Ministry of Education and Research, through project BetterView (FKZ 13N15827, 13N15830). T.H. acknowledges funding by the Deutsche Forschungsgemeinschaft (DFG, German Science Foundation), project number 540217954. Open Access funding is enabled and organized by Projekt DEAL.

## Author contributions

T.H., H.O., M.M. and G.H. contributed to the manuscript. H.O. conducted the imaging as well as the data analysis. The dSTORM imaging was conducted by H.O. and A. E.

## Funding

## Competing interests

G.H. is an employee of Excelitas PCO GmbH, the manufacturer of sCMOS cameras that were used in this work. The pco.edge 4.2 used in this work was purchased from Excelitas PCO GmbH, the other Excelitas PCO sCMOS cameras were provided to us free of charge by Excelitas PCO GmbH for the duration of the experiments. All other authors declare no competing conflicts of interest.
