## [Peer Review File · Nature Communications]

High sensitivity cameras can lower spatial resolution in high-resolution optical microscopyREVIEWER COMMENTS

Reviewer #1 (Remarks to the Author):

In this manuscript, the authors provide an interesting comparison of the modulation transfer functions of front-illuminated and back-illuminated cameras suitable for scientific imaging. They show that, despite their greater ability to detect photons, back-illuminated cameras may lead to a penalty in image resolution when used with the same pixel sizes. They demonstrate this experimentally for both wide-field imaging and structured illumination microscopy, and also show that these differences are not important when performing single molecule localisation microscopy.

The manuscript is well-written, the results are clear and the methodology appears sound. The fact that the MTF of a back-illuminated camera can lead to a large decrease in resolving ability in a typical microscope setup compared with more traditional front-illuminated cameras is perhaps not so surprising to camera experts, but will be to the vast majority of microscopists. As such, I support publication in Nature Communications.

However, there is one point that I suggest the authors consider, even though it does not affect any of the results presented: the effect of the MTF might be better interpreted as changing the field-of-view, not the resolution. The MTF of a back-illuminated camera will have a higher DC peak, as it collects more light. As such, there is room for using a higher magnification before the amount of light collected per pixel falls back down to the front-illuminated case (e.g. a 95% QE camera can have an extra magnification of 1.125x compared to a 75% QE camera). Given that the MTFs shown in the supplemental information are very similar when normalised, I suspect that the back-illuminated camera will perform just as well, if not better, when this extra magnification is taken into account.

As an aside, I'm surprised that the effect on the images is so pronounced considering the small differences in MTF. Perhaps it would be helpful to the reader to show calculated point spread functions from these MTFs, as this is something that the typical microscopist can more readily reason about.

== Minor comments ==

* Line 9: 'referred' -> 'preferred'

* Line 50: 'images' -> 'image'

* In the conclusion, I suggest not referring to electron microscopy as another area where this pixel-leakage could be a problem --- with new direct electron detectors they already have schemes for achieving (computational) super-resolution by exploiting the patterns of detected charges between neighbouring pixels.

* I suggest the authors emphasise that the images used have high signal levels, thereby reducing the contribution of read noise. This is particularly important when using FRC as a metric, as it can be 'confused' by any fixed pattern noise.

Reviewer #2 (Remarks to the Author):

In their paper "High sensitivity cameras can lower spatial resolution in high-resolution optical microscopy" by Ortkrass et al., the authors discuss the effect of the sensor MTF on high and super-resolution fluorescence imaging. The authors are comparing 4 different sensors, 2 front illuminated, 2 back illuminated from PCO.

The overall paper is well written and clear to understand. The authors are a bit overselling the article since the resolution loss is really noticeable in very precise cases so I would recommend to rephrase the introduction to clarify this point, making for instance a highlight on the space-bandwidth product of microscope objective and the need for field of view.

I have some remarks

- My major concern is regarding the comparison of camera from only one manufacturer with 3 sensor from Gsensor. A comparison with a camera with another sensor could very interesting, for instance the 11 μ m pixel back-illuminated from Teledyne (Kuro camera), or with an emCCD, with an adjusted magnification to be at the limit of Nyquist sampling. Indeed, it will magnify the interest of the paper, especially if the conclusions remain identical.

- It will be also very interesting to have this discussion for label-free imaging, such as incoherent DIC imaging which exhibit high frequencies in the formed imaged. It would be important to see the effect (or not) of the MTF degradation on such imaging modalities.
- It is also very important for me to see error bars, especially for the MTFs since the differences are tiny between each camera and since multiple beads were imaged and processed altogether.
- In fig.1 the images are, if I am right, interpolated. It doesn't help to understand that the effect of resolution loss is appearing at Nyquist sampling limit. I would like to see raw and interpolated images.
- When FRC results are presented, it would help the reader not familiar with FRC to indicate on the fig. what the '1/7' represents.

Reviewer #3 (Remarks to the Author):

Ortkraß et al present results on MTF characteristics of front vs back-illuminated image sensors. The results report that the MTF of camera sensors have an effect on the effective spatial resolution of a microscope. And this is particularly true in the common tube lens configurations found in laboratory wide-field microscopes today. My comments and questions, mostly regarding methodology and results:

"Back-illumination of images sensors does, however, come with a drawback: once photons are converted to photo-electrons in the doped silicon, they have to traverse a much longer path through the thinned silicon to reach the potential well of the pixel that collects them. This increases the possibility of electrons being scattered into neighboring pixels, an effect called pixel crosstalk.¹⁵"

Is this true regardless of exposure time? I.e. what happens in a high-photon regime, or a photon starved regime?

Given QE is an important factor, it would be good to characterize the MTF in a low-photon regime, as one is often found in biological imaging, particularly in living specimens where photons are precious.

And how much of the MTF is a function of the practical noise floor?

"To quantify the effect of the MTF on super-resolution fluorescence images we first measured the overall MTF of the microscope for all image sensors. Single 100 nm TetraSpeck (TS) beads (Thermo Fisher Scientific) were imaged in the center of the field-of-view (FOV) ten times each, utilizing almost the maximum dynamic range of each camera. This was repeated with several different beads. The contrast of the images was maximized by subtracting the background and the image stacks were Fourier-transformed, deconvolved with the lateral projected spherical bead shape, and averaged⁷."

Was deconvolution performed on the wide-field and SR methods? (It is unclear.)

The "algorithm" of explanation needs to be improved for clarity, since this is a key metric used by the authors.

How do the authors go from 3D data, to a 2D MTF? And what exactly is being averaged?

Is the "known support" the theoretically limited resolution or found experimentally?

Re. FRC:

As this is a key metric used, the metric should be explained in detail, with reference(s) cited (21 is cited later) if needed, and not assumed knowledge.

While I find the manuscript interesting and provides useful considerations for selecting camera sensors, I do not see a significant advance or improvement to wide-field detection more generally (with the difference between back vs front illumination being much less dramatic for SR and proper sampling). The take-home message appears to be that one

should sample slightly below the Nyquist limit to achieve the full theoretical resolution limit of the system, which is not new or surprising enough to warrant the manuscript suitable for Nature Communications.

Response to Reviewer's comments on the Nature Communications manuscript NCOMMS-24-09379, entitled " High sensitivity cameras can lower spatial resolution in high-resolution optical microscopy" by Henning Ortkrass, Marcel Müller, Anders Kokkvoll Engdahl, Gerhard Holst, and Thomas Huser

We would like to thank all reviewers for their valuable comments made in response to our manuscript. We have carefully revised the manuscript by taking all of these comments into consideration. In the following we provide a one-by-one response to each reviewer's comments. The reviewer's comments are printed in italicized and black font, whereas our response is shown in regular and blue font:

Reviewer 1:

In this manuscript, the authors provide an interesting comparison of the modulation transfer functions of front-illuminated and back-illuminated cameras suitable for scientific imaging. They show that, despite their greater ability to detect photons, back-illuminated cameras may lead to a penalty in image resolution when used with the same pixel sizes. They demonstrate this experimentally for both wide-field imaging and structured illumination microscopy, and also show that these differences are not important when performing single molecule localisation microscopy.

The manuscript is well-written, the results are clear and the methodology appears sound. The fact that the MTF of a back-illuminated camera can lead to a large decrease in resolving ability in a typical microscope setup compared with more traditional front-illuminated cameras is perhaps not so surprising to camera experts, but will be to the vast majority of microscopists. As such, I support publication in Nature Communications.

However, there is one point that I suggest the authors consider, even though it does not affect any of the results presented: the effect of the MTF might be better interpreted as changing the field-of-view, not the resolution. The MTF of a back-illuminated camera will have a higher DC peak, as it collects more light. As such, there is room for using a higher magnification before the amount of light collected per pixel falls back down to the front-illuminated case (e.g. a 95% QE camera can have an extra magnification of 1.125x compared to a 75% QE camera). Given that the MTFs shown in the supplemental information are very similar when normalised, I suspect that the back-illuminated camera will perform just as well, if not better, when this extra magnification is taken into account.

We thank the reviewer for this comment. The reviewer is certainly correct that in the case of a back-illuminated camera, utilizing a slightly higher magnification of the optical system could potentially compensate the lower spatial resolution obtained at otherwise equal magnification. At present, however, we could not implement this request in our experiments, because it would have meant that we would have to change the optical system, which then creates the problem that the MTF will also change in response to the optical changes and it will become very difficult to distinguish camera-dependent vs. optics-dependent changes in the MTF. Also, most commercial microscopes don't provide the ability to seamlessly change the magnification. And, lastly, a change in magnification will also result in an overall higher noise, because now the information from additional pixels will be contributed to the MTF. All of these factors highly complicate the characterization of changes to the optical system. Thus, in response to this comment, we added a sentence to the conclusions of the paper: "This effect could potentially be compensated by a change in magnification based on the sensor type, but this is often difficult to realize in practice."

Furthermore, we would like to point out that in order to provide a fair comparison between sensor types we divide the image data 50:50 between the different cameras and also swap cameras between these two paths. Thus, a simple normalization will not change the performance of either camera. Although the MTFs are normalized, the resolution is still significantly worse for the back-illuminated camera although the image data are not being normalized.

As an aside, I'm surprised that the effect on the images is so pronounced considering the small differences in MTF. Perhaps it would be helpful to the reader to show calculated point spread functions from these MTFs, as this is something that the typical microscopist can more readily reason about.

We fully agree with the reviewer and have provided point spread functions (PSFs) in addition to the MTFs in the supplemental information.

== Minor comments ==

* Line 9: 'referred' -> 'preferred'

* Line 50: 'images' -> 'image'

We have corrected these typos in the revised manuscript.

** In the conclusion, I suggest not referring to electron microscopy as another area where this pixel-leakage could be a problem --- with new direct electron detectors they already have schemes for achieving (computational) super-resolution by exploiting the patterns of detected charges between neighbouring pixels.*

We thank the reviewer for this comment and have removed the mentioning of electron microscopy in the revised manuscript.

** I suggest the authors emphasise that the images used have high signal levels, thereby reducing the contribution of read noise. This is particularly important when using FRC as a metric, as it can be 'confused' by any fixed pattern noise.*

We thank the reviewer for this comment. In the revised manuscript, in response to reviewer 1, but also reviewers 2 and 3, we have included additional MTF and FRC data acquired in the high photon-count regime in the revised supplemental information. These data demonstrate that the read noise of the different cameras has a rather minor effect on the overall spatial resolution which can be achieved by utilizing the cameras in otherwise identical optical imaging systems. Nonetheless, we also added a statement to the main manuscript text to emphasize that all image data (except for the α STORM) data were acquired with high signal levels.

To satisfy this request we added several sentences to the first section of the "Results and Discussion" chapter:

In the second paragraph of this chapter, we added the following sentences:

"MTF data acquired by imaging fluorescent beads are considered as data acquired in a low-photon count regime. To further explore the influence of measurements taken in a high-photon count regime, noise, and alternative incoherent light sources on the overall MTF we also acquired MTF data by illuminating a 200 nm diameter hole in an otherwise opaque aluminum film. Here, a 200 nm diameter hole was milled by focused ion-beam milling into a 100 nm thick

aluminum layer on a 170 μm thick cover glass. The hole was positioned in the center of the field of view and then illuminated with a white light LED from behind."

In the third paragraph of this chapter, we also added the statement:

"It should also be noted that the samples imaged in widefield fluorescence and SR-SIM modalities all exhibited high fluorescence signal levels."

Reviewer 2:

In their paper "High sensitivity cameras can lower spatial resolution in high-resolution optical microscopy" by Ortkrass et al., the authors discuss the effect of the sensor MTF on high and super-resolution fluorescence imaging. The authors are comparing 4 different sensors, 2 front illuminated, 2 back illuminated from PCO. The overall paper is well written and clear to understand. The authors are a bit overselling the article since the resolution loss is really noticeable in very precise cases so I would recommend to rephrase the introduction to clarify this point, making for instance a highlight on the space-bandwidth product of microscope objective and the need for field of view.

We thank reviewer 2 for this comment. Since the introduction of our paper does not discuss this effect, we believe that the reviewer wants us to rephrase the abstract. In response to this comment we have added an additional sentence (and rephrase the sentences before and after this sentence), to accommodate the reviewer's request. The rephrased part of the abstract now reads:

"... Here, we show that the drive to use ever more sensitive, photon-counting image sensors in cameras can, however, have detrimental effects on the spatial resolution of the resulting images. This is particularly noticeable in applications that demand a high space-bandwidth product, where the image magnification is close to the Nyquist sampling limit of the sensor. Most scientists will often select image sensors based on parameters such as pixel size, quantum efficiency, signal-to-noise performance, dynamic range, and frame rate of the sensor. A parameter that is, however, typically overlooked is the sensor's modulation transfer function (MTF). ..."

- My major concern is regarding the comparison of camera from only one manufacturer with 3 sensor from Gsensor. A comparison with a camera with another sensor could very interesting, for instance the 11 μm pixel back-illuminated from Teledyne (Kuro camera), or with an emCCD, with an adjusted magnification to be at the limit of Nyquist sampling. Indeed, it will magnify the interest of the paper, especially if the conclusions remain identical.

We thank the reviewer for this comment and this advice. In the revised manuscript we have now included additional data obtained with a sCMOS camera made by Andor Technology (sensor produce by Fairchild) and a first generation Hamamatsu Orca Flash. Both are front-illuminated cameras, but with a pixel size consistent to that of the PCO cameras that we tested in the original manuscript. It is however, unfortunately, very difficult, if not impossible, to test the cameras that the reviewer suggested. The reason is that e.g. the Teledyne camera is only available as a back-illuminated version and a comparable front-illuminated camera with an 11 μm pixel size does not exist. Similarly, the current generation of emCCD cameras (pixel size 16 μm) are also only available as back-illuminated cameras and not as front-illuminated cameras. Because of the missing reference system for either one of these cameras it is very difficult to determine specific effects of the camera MTF, which are not affected by the MTF of the optical system. For the same reason, we also cannot compare these cameras to the smaller pixel size BI PCO cameras, because different magnifications and, thus, different optical components are

required to obtain comparable image data. Furthermore, the noise characteristics of emCCDs is very different from that of sCMOS cameras, which would further complicate a fair comparison. Please note that we only used two BI cameras in the original manuscript in order to demonstrate that two cameras with the same pixel size, same manufacturer, but different manufacturing processes result in different MTFs. In the revised manuscript we included additional data (MTFs, PSFs, as well as image data and FRC data) taken with the Andor and Hamamatsu sCMOS cameras in the supplemental information.

- It will be also very interesting to have this discussion for label-free imaging, such as incoherent DIC imaging which exhibit high frequencies in the formed imaged. It would be important to see the effect (or not) of the MTF degradation on such imaging modalities.

We very much appreciate this comment and suggestion made by reviewer #2. Unfortunately, DIC imaging is a modality that we do not utilize very often and we are, thus, unable to collect data with this modality within a reasonable timeframe. We also wonder if this is really necessary, because DIC imaging typically provides a significantly higher signal level than fluorescence microscopy, and sCMOS cameras are not typically used for this application, but rather much more inexpensive cameras. Still, in order to address this comment in a reasonable manner, we retook MTF data with all camera systems by imaging white light penetrating through a focused ion-beam milled 200 nm diameter hole in an aluminum film. This should simulate the white light approach used in DIC imaging, it provides contrast on the nanoscale; and we have included the additional MTF data in the supplemental information of the revised manuscript.

- It is also very important for me to see error bars, especially for the MTFs since the differences are tiny between each camera and since multiple beads were imaged and processed altogether.

We thank the reviewer for making this point. Indeed, all plots provided do contain error bars, but the error bars are often extremely small and can barely be seen in the current plots. We would like to direct the reviewer to Suppl. Fig. 1 and the insets to Suppl. Fig. 3(a and b). In Suppl. Fig. 1 the error bars are actually well visible as lighter color areas surrounding the middle line of the MTF. In the insets to Suppl. Fig. 3, the lighter color areas are also visible, although they can only be identified if one zooms in very close to the MTF curves.

- In fig.1 the images are, if I am right, interpolated. It doesn't help to understand that the effect of resolution loss is appearing at Nyquist sampling limit. I would like to see raw and interpolated images.

We agree with the reviewer and, in addition to the interpolated images shown in Fig. 1 in the main text, we have now also added a replicate of Fig.1 showing raw, non-interpolated images as Suppl. Fig. 5.

- When FRC results are presented, it would help the reader not familiar with FRC to indicate on the fig. what the '1/7' represents.

An additional sentence to explain how FRC data are processed from image data has been added to the supplemental information file in the section labeled "Image resolution measurement". The reference that refers to the original paper where FRC was introduced to the optical microscopy community is now also being cited earlier in the text. The 1/7 threshold which we used consistently throughout the manuscript in order to provide quantitative values of the spatial resolution for each image is shown in all FRC plots by dashed lines.

Reviewer 3:

Ortkraß et al present results on MTF characteristics of front vs back-illuminated image sensors. The results report that the MTF of camera sensors have an effect on the effective spatial resolution of a microscope. And this is particularly true in the common tube lens configurations found in laboratory wide-field microscopes today. My comments and questions, mostly regarding methodology and results:

"Back-illumination of images sensors does, however, come with a drawback: once photons are converted to photo-electrons in the doped silicon, they have to traverse a much longer path through the thinned silicon to reach the potential well of the pixel that collects them. This increases the possibility of electrons being scattered into neighboring pixels, an effect called pixel crosstalk."

Is this true regardless of exposure time? I.e. what happens in a high-photon regime, or a photon starved regime? Given QE is an important factor, it would be good to characterize the MTF in a low-photon regime, as one is often found in biological imaging, particularly in living specimens where photons are precious. And how much of the MTF is a function of the practical noise floor?

We thank the reviewer for this comment. As already mentioned earlier, we have retaken the MTF data by using a different approach. The original MTF data were acquired by positioning a single fluorescent beads in the center of the field of view, which can be considered the "photon starved" regime. In the revised manuscript we have also included MTF data that we obtained by positioning a single 200 nm hole in an aluminum film in the center of the field of view. The hole was then illuminated by bright light from a white LED and the transmitted light was collected with a low magnification to reduce the influence of the optics MTF. This allowed to image the bright hole with a resolution of 188 nm onto a single pixel with a projected pixel size of 550 nm and to measure the camera MTF. Here, we can vary the amount of light that penetrates this hole based on the brightness of the LED and this also allows us to create the conditions considered as the high-photon count regime. The additional MTFs are shown in Suppl. Fig. 1 in the supplemental information of the revised manuscript. The results of these different MTF measurements are, however, very similar to the original data. We have further analyzed this in Suppl. Fig. 2, where MTFs are calculated for camera C4 in different photon count regimes. Without subtracting the noise floor, the MTFs appear to vary significantly, but if the noise floor is subtracted before normalizing the MTF, all MTFs are very similar. This shows that the MTFs that we collected in the high-photon count regime are minimally impacted by the noise floor. We also calculated FRC plots for both acquisition schemes. Both sets of data demonstrate that in neither case the spatial resolution of the images is significantly affected by the noise floor.

The "algorithm" of explanation needs to be improved for clarity, since this is a key metric used by the authors. How do the authors go from 3D data, to a 2D MTF? And what exactly is being averaged? Is the "known support" the theoretically limited resolution or found experimentally?

We believe that our reference to "deconvolution" might have confused the reviewer. All image data collected for this manuscript are, indeed, 2D image data. Deconvolution only refers to the widefield fluorescence images, where the raw images have been deconvolved by the point spread function of the microscope in order to achieve the highest spatial resolution available with widefield microscopy. "Averaging" only applies to the calculation of the MTF data. Here, the 2D images are azimuthally averaged in order to obtain the MTF plots. We have clarified this in the main manuscript by adding the sentence " The two-dimensional MTF was azimuthally averaged and set to zero outside its (theoretically calculated) known support." By adding this

sentence, we have now also clarified that the microscopes support was determined theoretically.

Re. FRC: As this is a key metric used, the metric should be explained in detail, with reference(s) cited (21 is cited later) if needed, and not assumed knowledge.

We thank the reviewer for this comment. In the revised manuscript, the reference to the original paper was moved up to the first mention of FRC. Furthermore, we added the following sentences in order to briefly explain FRC: " FRC is a method originally introduced in electron microscopy in order to measure the correlation between two images at different spatial frequencies. For this purpose, two images of the same sample have to be acquired and are then correlated against each other. If the correlation coefficient drops below $1/7$ (0.14), the signal is dominated by noise and the resolution cutoff is reached. We use this value throughout the paper to compare the spatial resolution that can be reached with different image sensors with otherwise identical optical systems. "

While I find the manuscript interesting and provides useful considerations for selecting camera sensors, I do not see a significant advance or improvement to wide-field detection more generally (with the difference between back vs front illumination being much less dramatic for SR and proper sampling). The take-home message appears to be that one should sample slightly below the Nyquist limit to achieve the full theoretical resolution limit of the system, which is not new or surprising enough to warrant the manuscript suitable for Nature Communications.

We appreciate this comment made by reviewer #3, but beg to differ. The majority of biologists are not typically aware of the Nyquist sampling limit and its implications and will typically try to improve the sensitivity of their imaging system rather than paying attention to its implications on spatial resolution. Based on our comparison of different sensor types and the resulting image data we believe that it becomes very clear that even when sampling slightly above the Nyquist limit the spatial resolution is still significantly affected and can differ by up to 10% regardless of the sensor size and pixel size. We believe that Figure 1 in the manuscript demonstrates this effect very clearly. One has to acquire data significantly above the Nyquist sampling limits (depending on the application by up to 30 - 50% above the N limit) in order to avoid this impact on spatial resolution. We have added this statement to the conclusions of the manuscript in order to emphasize this point.

REVIEWERS' COMMENTS

Reviewer #1 (Remarks to the Author):

I commend the authors for their efforts, which have dealt with all the points I raised in my previous review. I strongly recommend publication in Nature Communications.

Reviewer #1 (Remarks on code availability):

The code was not available through the provided Figshare DOI link.

Reviewer #2 (Remarks to the Author):

The authors addressed most of my comment of my first review and the paper is less over-selling the results now. The FTM effects on images remain still limited (hopefully!) but they are well described and characterized.

Reviewer #2 (Remarks on code availability):

Link not accessible

Reviewer #3 (Remarks to the Author):

The authors addressed my concerns and the manuscript has improved. I also appreciate the data, software and code sharing. Kudos to the authors.

That said, as I stated in my previous review, I still do not think this is a fundamental leap in technology development to warrant publication in Nature Comm, though I do think it is informative and interesting for anyone selecting a camera for a camera-based microscope.